# Rethinking the Impact of Social Media Exposure and Source Credibility on the Social Amplification of Risk and Public Engagement During the COVID-19 Pandemic

**Longfei Li [1] and Ran Feng [2,*]**

[1]   School of Journalism and Communication, Minzu University of China, Beijing 100081, China; longfei_sjtu@163.com
[2]   Faculty of Humanities and Social Sciences, City University of Macau, Macau 999078, China
*   Correspondence: rfeng@cityu.edu.mo

**Abstract:** Promoting public engagement through social media has always been a core issue in risk communication studies. Based on the Social Amplification of Risk Framework (SARF), this study conducts an online survey in China (N = 908) and constructs a moderated mediation model. Using bootstrapped moderated mediation analysis, this study examines the relationships among social media exposure to pandemic information, risk perception, source credibility, and public engagement on social media at the early stages of the pandemic. The results demonstrate a positive relationship between social media exposure and public engagement, which can be mediated by risk perception. The relationship between social media exposure and public engagement via risk perception is moderated by source credibility. The higher perceived credibility of official and interpersonal sources undermines risk perception, but also hinders public engagement in the crisis. The moderating effect of professional source credibility is not significant. This study has expanded the SARF and has contributed to promoting risk communication strategies from the perspective of risk information processing.

**Keywords:** media exposure; source credibility; risk perception; public engagement; COVID-19 pandemic

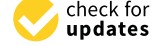

## 1. Introduction

We live in a mediated world, a risk society, and a post-trust space. When the world fights against the pandemic, it is facing an "infodemic," which means that some risk events that are assessed by technical experts as relatively small can attract strong public attention and have substantial impacts on society, while some others often go unnoticed [1]. With the development of digitalization, social media can quickly disseminate scientific information, but also amplify harmful messages. Especially at the beginning of the pandemic, an infodemic may intensify or lengthen outbreaks when people are unsure about what they need to do to protect their health and the people around them. Given the central role of social media as a key tool for risk communication, scholars have synthesized the research landscape within the first year of the COVID-19 outbreak, covering topics such as public attitudes, mental health, media effects, and governmental responses. As social media increasingly constitutes a platformized environment for risk communication, there is an urgent need for further nuanced investigation into its role in mobilizing, organizing, and shaping health-related risk behaviors [2].

The influence of media exposure on public attitudes and behaviors has long been studied in the literature on persuasion and health communication [3–6]. However, the dynamic and nonlinear nature of pandemic development, particularly during its initial outbreak, has not been adequately captured by existing studies. This study aims to deepen this understanding by investigating the early-stage dynamics of risk communication and public engagement, thereby contributing to a more comprehensive understanding of how infodemics evolve and how communication strategies might be improved in future public health crises. The Social Amplification of Risk Framework (SARF) proposes that media, as a social amplification station for risk communication, can transmit risk information signals, amplify or weaken people's risk perception, and thus restrain or trigger risk actions. Shifts in the media landscape, such as the rise of short-video platforms and the decline in public trust toward mainstream media, have underscored the need to revisit risk communication mechanisms within an increasingly fragmented and attention-driven digital ecosystem. For one thing, social media platforms have the characteristic of real-time information dissemination, allowing for the rapid release of pandemic updates, protective guidelines, and risk alerts, helping the public stay informed and respond promptly [7]. More importantly, social media also serves as an interactive communication platform where people can engage in real-time discussions, leave comments, and share their experiences, thereby promoting mutual assistance and cooperation [8]. Public active engagement on social media during a crisis is of critical relevance [9]. Furthermore, some researchers argue that source credibility is a complex attitude variable that can influence risk perception and willingness to participate [10]. It is therefore important to empirically investigate the following research questions.

RQ1: Does social media exposure amplify or reduce risk perception, and how does it affect people's response to risk during the early stage of the pandemic?

RQ2: Does the interaction between social media exposure and source credibility influence risk perception and risk engagement?

Due to the evolving nature of health risks, previous studies have mostly focused on the general population and lacked additional attention to the center of the early pandemic. Conducting an in-depth study of the characteristics of early risk sources can help reduce or prevent the occurrence of secondary disasters. Thus, this study focuses on residents in Hubei province during the outbreak in China, with Wuhan being the capital city. Based on the SARF, this study investigates how social media exposure and source credibility affect people's risk perception and public engagement.

## 2. Literature Review

### 2.1. Social Media and the Theory of SARF

In the risk analysis field, the understanding of risk has evolved significantly over the past three decades, from being viewed merely as a technical hazard involving mechanical loss to encompassing psychological measurement paradigms and risk communication processes. Throughout this evolution, the Social Amplification of Risk Framework (SARF) serves as a foundational theoretical tool for analyzing how media and social actors shape public responses to risk. It has also proven instrumental in facilitating risk communication and governance among regulatory agencies, experts, and the public. The model proposes two main mechanisms: risk information transmission and societal response [11]. The former posits that prolonged exposure to media can cultivate users' perceptions, aligning them more closely with mediated representations rather than objective conditions [12]. This suggests that media channels, mainly traditional media such as newspapers, radio, and television, act as amplification stations that can intensify or attenuate public risk awareness and subsequent behavioral responses.

However, the rise of social media poses new challenges to the SARF. A study in 2011 was among the first to call for a reconsideration and reassessment of the SARF model considering the digital media environment [13]. Researchers further emphasized that social media should be conceptualized as an independent amplification station [14]. First, the open nature of social media facilitates active user participation and expands public discourse on risk and its communication. This creates a compelling "playground" for risk researchers to examine how individuals perceive and respond to risk events. Second, algorithmic filtering on social media platforms tailors content to user preferences, often leading to the formation of "echo chambers" where risk information circulates primarily within like-minded communities. As a result, risk communication is no longer merely a linear or socially driven amplification—it can be algorithmically ranked, emotionally triggered, and dominated by sensational content. These platform-specific dynamics reflect a novel form of risk amplification that departs from traditional models and call for renewed investigation into risk engagement behaviors in the contemporary digital media ecosystem. For instance, within the context of social media, SARF is often invoked to explain why certain health-related events, despite receiving moderate technical risk assessments, nonetheless trigger disproportionate public concern or behavioral shifts [15]. Media exposure has been widely used as a predictor variable in studies measuring its catalytic impact on public risk perception during health crises [16]. Thus, SARF provides a useful lens for analyzing the links between social media exposure, risk perception, and public engagement during the COVID-19 pandemic.

Furthermore, in the context of digital risk communication, researchers have suggested that rather than focusing on the negative impact of social media, it is better to focus on its potential to maintain and cultivate trust, which is an important criterion for people's judgment and decision-making [17]. Indeed, the relationship between social media exposure and public engagement may be disturbed, considering the different perceived credibility of information sources. Scholars generally divide media credibility into source credibility and content credibility [18]. Source credibility in health communication refers to a measure of characteristics that make the information more or less believable [19]. In risk communication events in social media, different public opinion fields are often formed by official, professional, and personal sources. They compete for attention and significance of risk event construction. Authoritative information comes from the government and official media, representing the will of authority. It has the switching power to control the dissemination of information and the ability to project thoughts, resources, and human resources during the pandemic onto the existing network [20]. In the aspect of risk warning, the information from official public health generates more powerful risk perception and risk defense behaviors for citizens than unofficial information [21]. Experts and scholars from professional medical agencies have begun to play an important role in public health crises [22,23]. In contrast to the information from authoritative institutions, ordinary people can also make use of their social media to communicate with their family, neighbors, and friends, through which interpersonal communication could tacitly influence risk behaviors. For example, during the COVID-19 pandemic, US adults, despite their exposure to the Internet, experienced a profound impact on risk perception and protective behavior through credibility statistics about the health information sources of those around them [24]. However, the content shared on social media platforms is multifaceted. These platforms possess the capability to programmatically distribute information, allowing for the selective linking of ideas and resources, including unverified misinformation and rumors. Therefore, the application of the SARF model needs to be further improved to discuss how the interaction effect of social media source credibility and social media exposure in public health crises on risk perception and public engagement.

COVID-19 began as an unknown disease, and its uncertainty has brought cognitive conflict all over the world. However, throughout the current research on the pandemic, we neglected to study social media engagement with risk in hard-hit areas during the early stage of the pandemic, despite the residents being the first group to face the unknown disease and use social media to help. The relevant information obtained through social networks during a crisis is crucial to people's risk identification [25]. Individuals directly affected by the crisis may provide more vivid and detailed descriptions of various aspects of the emergency, and the dissemination and discussion of risk information can better identify the danger of the situation for others [26]. Based on the above discussion, this study aims to explore how social media, as a risk amplification station, affected risk perception and public engagement in the early stage of the pandemic in China, and the role played by source credibility.

### 2.2. Mediating Role of Risk Perception Between Social Media Exposure and Public Engagement

Public engagement in health risks describes the audience's interest in and willingness to interact with risk response [27]. For example, we have observed that some residents would use social media to share information, leave comments, write diaries, etc., to participate in risk communication activities during the initial stage of the epidemic. Multiple studies have shown the direct impact of social media exposure on promoting health and risk engagement, with a range of advantages, including the rapid dissemination of information [28], allowing for interaction with target audiences, and establishing conversational relationships [29]. A meta-analysis of 2040 studies across eight databases showed a strong link between online social networking and behavioral change [30]. Social networks provide the public with more health risk information and knowledge, which can potentially stimulate engagement in risky behaviors.

Nevertheless, the underlying mechanisms of how social media information influences public engagement have not been fully understood. Increasing evidence suggests that the impact of social media information on health risk actions may be related to personal perception of health risk changes, which is a key factor in behavior change [31]. Health risk perception is a subjective psychological structure that involves individuals' cognition, assessment, and judgment of health threats [32]. SARF theory holds that media, as a social amplification station, can trigger risk perception. The key to the question is not whether an individual has direct risk experience, but whether media can amplify or weaken risk perception [33]. The functions of media are mainly reflected in providing information about diseases, acting as an intermediary between the public and policymakers of disease prevention, and monitoring the interests of individuals and institutions [34]. A study found that the increase in mortality during the COVID-19 pandemic has led to a growing awareness of its risks [35]. Negative emotions are more likely to appear on social media than positive emotions, thus increasing the public risk perception of infectious diseases [36]. Therefore, the increase in exposure to risk information through social media may induce a higher perception of risks. In addition, risk perception can serve as a powerful motivator for risk response [37]. Multiple studies on the COVID-19 pandemic have shown there is a strong correlation between using risk perception to predict public engagement in health protection [38]. Thus, we propose the first hypothesis:

**H1:** *Risk perception of the pandemic serves as a mediating variable between social media exposure and online public engagement.*

### 2.3. Moderating Role of Source Credibility Between Social Media Exposure and Risk Perception

Although some studies have shown that exposure to social media information during health risk incidents can contribute to public engagement, some researchers suggest that

there is still a gap between information exposure and public action. This is because the perception of information source credibility can influence the relationship between media exposure, risk perception, and public participation [39].

Previous research has found that the source of information can influence the persuasiveness effect [40]. The impact of information sources on the recipient's understanding of the information depends on factors such as the identity of the communicator, communication mechanisms, recipient characteristics, and information context. Different information sources possess varying levels of expertise, competence, honesty, and helpfulness, which in turn influence their credibility in the minds of information receivers [41]. For example, the rapid development of social media platforms has intensified concerns about false information, making the public more interested in source credibility [42]. Despite the increasing internet use to search for risk information, news on the internet is inaccurate and misleading because it is not fact-checked, and many reporters are not well-trained [43]. Some researchers suggest that the public has a high degree of trust in the crisis information provided by authoritative journalists, believing that the identity and background of the news source directly affect information credibility. Due to the public recognition of experts' authority, medical experts (virologists, epidemiologists, public health professionals, and statisticians) played a role in recommending policies to address the spread of coronavirus [44]. Cultivation theory also points out that although TV exposure can effectively construct people's beliefs, attitudes, and values, when information source monitoring plays a role as a moderating variable, the cultivation effect brought by media exposure will weaken, which is called source discounting [45].

Several empirical studies have shown that source credibility can moderate the relationship between information exposure and risk perception [37]. Prior research has shown that risk perception is closely related to information source credibility, which can affect information search and processing [46]. During the Netherlands' influenza A (H1N1) pandemic, most respondents wanted to receive information about infection prevention from municipal health services, health care providers, and professional media [47]. People who lack personal knowledge of hazards always rely on authoritative social trust to evaluate risks and benefits [48]. Experimental research shows that the medium itself is more important than the message in crisis communication, and media with good reputations acquire public trust and provoke actions [41].

At the beginning of the pandemic in China, we observed that the public obtained health information from various sources through social media. There were three main sources: official, professional, and personal sources. Especially in Wuhan, facing the uncertainty of coronavirus, media institutions showed significant differences in their attitude toward risk representation. For example, official media had blind spots in cognition and reporting, failing to highlight the severity of the pandemic, which may also have weakened the risk perception of people who were concerned about the credibility of official media. Therefore, we argue that the credibility of different channels would moderate the impact of social media exposure on risk perception. In line with this, the following hypothesis is formulated:

**H2:** *Information source credibility moderates the relationship between social media exposure and risk perception of the pandemic in such a way that the effect is stronger when information source credibility is higher.*

Building upon prior studies and extending the Social Amplification of Risk Framework (SARF) from the perspective of psychological perception mechanisms, the refined research model is illustrated in Figure 1.

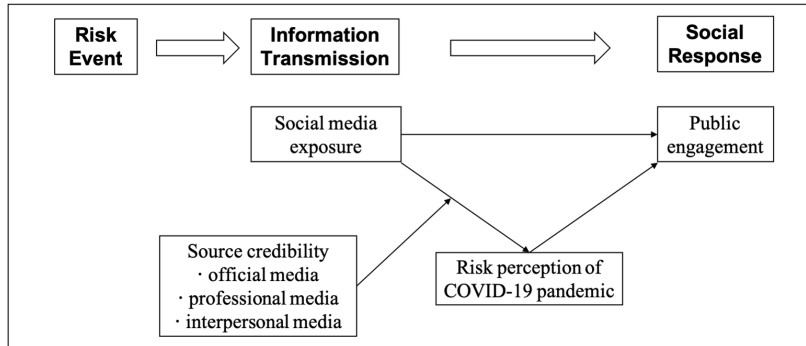

**Figure 1.** Refined theoretical model of SARF.

## 3. Methods

### 3.1. Participants

This survey was conducted online in March 2020. The participants were living in Hubei province, which was the worst-hit area at the initial stage of the pandemic. They were recruited through convenience sampling from the Tencent Questionnaire Service Platform (equivalent to MTurk). Each participant completed a self-report survey together with a consent form. The consent form stated the purpose of the research and the rights of the participants and offered a monetary incentive (equivalent to USD 2). At the end of the data collection, 915 completed responses and 908 valid questionnaires were obtained. The sample comprised 628 male (69.2%) and 280 (30.8%) female respondents, with an age distribution of 6.2 percent below 16, 90.1 percent from 18 to 45, and 3.7 percent above 46. In total, 41.6 percent of participants were living in big cities, 41.2 percent in medium-sized cities, and 17.2 percent dwelling in rural areas. The sample distribution of respondents is consistent with the population distribution of Hubei. A Supplementary File S1 for the questionnaire is available as noted on page 12.

### 3.2. Measures

#### 3.2.1. Social Media Exposure

To measure social media exposure to pandemic information, this study modified the scale of a previous study [49], including the most used social media apps (WeChat, Weibo, Toutiao, TikTok, Douban). The participants rated 5 statements based on the degree to which they felt each statement described them, responding to a 5-point scale (1 = never, 5 = very frequently). A composite measure formed by averaging participants' responses to these items assessed the level of social media use (Cronbach's $\alpha$ = 0.82).

#### 3.2.2. Risk Perception

For the measurement of risk perception during the pandemic, we adopted the risk perception scale [50], which has been widely used in the risk and health communication literature. Participants rated four statements regarding the degree to which each statement described their threat perception of COVID-19 (1 = strongly disagree to 5 = strongly agree). The CFA results indicated that all factor loadings were significant with a good model fit (NFI = 0.97, CFI = 0.97, GFI = 0.98, RMSEA = 0.14). Standardized loadings ranged from 0.71 to 0.79. The reliability of the measure was good (Cronbach's $\alpha$ = 0.84).

#### 3.2.3. Source Credibility

To assess the source credibility of pandemic-related information on social media, this study refers to a scale developed in previous research [51]. Participants rated their level of trust toward three distinct types of information sources: (1) official media (e.g., government-affiliated and mainstream institutional media), (2) professional media (e.g., expert-led medical

or public health media), and (3) interpersonal media (e.g., personal accounts such as WeChat posts from friends and family). Each source type includes five items, rated on a 5-point Likert scale (1 = no trust at all, 5 = complete trust). This study treats source credibility as a multidimensional construct, reflecting differentiated trust across three types of media sources. This approach allows for a more nuanced understanding of how trust in various sources influences individuals' risk perception and engagement behavior.

### 3.2.4. Public Engagement

According to a related study [9], we used three items to measure public engagement in dealing with risk using social media (1 = never, 5 = very frequently), including sharing pandemic information to people in need; participating in the online pandemic discussion; presenting self-disclosure behaviors such as commenting, and writing pandemic diaries. Then, we calculated the average as the score of public engagement (Cronbach's $\alpha$ = 0.79).

### 3.2.5. Controlling Variables

The statistical analysis measured and controlled for demographic variables that might affect the dependent variables, including gender, age, income, level of education, and residence. In addition, mass media exposure should be controlled to measure the effect of social media exposure. Table 1 reports the descriptive statistics of all the variables.

**Table 1.** Descriptive statistics of variables (N = 908).

| Variable | Mean | SD | Min | Max |
|---|---|---|---|---|
| Public engagement | 3.58 | 1.13 | 1 | 5 |
| Risk perception | 3.66 | 1.08 | 1 | 5 |
| Source credibility | | | | |
| Official media | 3.58 | 1.23 | 1 | 5 |
| Professional media | 3.66 | 1.29 | 1 | 5 |
| Interpersonal media | 3.19 | 1.24 | 1 | 5 |
| Social media exposure | 3.55 | 1.04 | 1 | 5 |
| Mass media exposure | 3.11 | 1.20 | 1 | 5 |
| Community media exposure | 3.27 | 1.48 | 1 | 5 |
| Interpersonal media exposure | 3.50 | 1.37 | 1 | 5 |
| Gender (1 = male) | 0.69 | 0.46 | 0 | 1 |
| Age range | 2.82 | 0.93 | 1 | 5 |
| Family income | 2.32 | 0.82 | 1 | 4 |
| Education (1 = primary school and below) | 3.26 | 0.99 | 1 | 5 |
| Residence (1 = urban) | 0.83 | 0.38 | 0 | 1 |

## 4. Results

To test the mediating role of pandemic risk perception between social media exposure and public engagement, a set of hierarchical regressions was conducted. The mediation hypothesis (H1) was examined with the procedure proposed by Preacher and Hayes [52], which depicted three steps to deal with a mediation model: first, the independent variable should be significantly correlated with the mediator; second, after controlling for the independent variable, the correlation coefficient between the mediator and the dependent variable should be significant; third, the indirect effect between the independent variable and the dependent variable via the mediator should be significant in the bootstrapping test [53].

As we can see from Table 2, Model 1 and Model 3 are initial models that include the control variables in the regression equation. The results show that compared with urban residents, the risk perception for rural residents was significantly higher ($\beta$ = −0.108, $p < 0.001$), and exposure to mass media and interpersonal networks could also significantly

enhances risk perception ($\beta$ = 0.227, $p$ < 0.001; $\beta$ = 0.219, $p$ < 0.001) and willingness to deal with pandemic risks ($\beta$ = 0.218, $p$ < 0.001; $\beta$ = 0.268, $p$ < 0.001).

**Table 2.** Regression results for predicting risk perception and public engagement.

| Dependent Variables | Risk Perception | | Public Engagement | | |
|---|---|---|---|---|---|
| | Model 1 | Model 2 | Model 3 | Model 4 | Model 5 |
| Gender | −0.062 * | −0.051 | −0.069 * | −0.056 | −0.031 |
| Age range | 0.069 * | 0.052 | 0.024 | 0.004 | −0.021 |
| Family income | 0.058 | 0.050 | 0.006 | −0.004 | −0.028 |
| Education | 0.060 | 0.037 | 0.129 *** | 0.100 ** | 0.082 ** |
| Residence | −0.108 *** | −0.077 * | −0.045 | −0.008 | 0.029 |
| Mass media exposure | 0.227 *** | 0.205 *** | 0.218 *** | 0.192 *** | 0.093 ** |
| Community media exposure | 0.052 | 0.051 | 0.073 * | 0.072 * | 0.047 |
| Interpersonal media exposure | 0.219 *** | 0.199 *** | 0.268 *** | 0.244 *** | 0.148 *** |
| Social media exposure | | 0.182 *** | | 0.220 *** | 0.133 *** |
| Risk perception | | | | | 0.482 *** |
| $R^2$ | | | 0.2484 | 0.2914 | 0.4673 |
| N | 908 | 908 | 908 | 908 | 908 |

Notes: 1. * $p$ < 0.05, ** $p$ < 0.01, *** $p$ < 0.001 2. Use standardized coefficient.

In Model 2 and Model 4, social media exposure was added to the regression model based on all control variables. The results indicate that social media exposure had a positive impact on risk perception ($\beta$ = 0.182, $p$ < 0.001), and social media exposure helps to enhance public engagement ($\beta$ = 0.220, $p$ < 0.001). In Model 5, risk perception, the mediating variable, was added to the regression equation, and the results show that risk perception positively affected public engagement ($\beta$ = 0.482, $p$ < 0.001). The above results provided preliminary support for the mediating role of risk perception between social media exposure and public engagement. To further examine the mediating effect, we followed the recommendations of Preacher and Hayes and adopted statistical software developed by Hayes with a bias-corrected bootstrapping procedure [53]. The bootstrapping (N = 908, bootstrap samples = 5000) results demonstrated the mediating role of risk perception between social media exposure and public engagement ($\beta$ = 0.085, 95%CI = [0.058, 0.116]). Thus, H1 is supported.

To examine whether source credibility moderates the indirect effect of social media exposure on public engagement via risk perception during the early stage of COVID-19, this study employs a moderated mediation model using Hayes' PROCESS macro (Model 7). Bootstrapping with 5000 resamples is applied to test the significance of indirect effects across different levels of source credibility. In this model, social media exposure serves as the independent variable, risk perception as the mediator, and public engagement as the dependent variable, while the perceived credibility of official, professional, and interpersonal sources is treated as a moderator on the first-stage path. The visual representation of the conditional process model is illustrated in Figures 2–4 to facilitate the interpretation of the interaction effects. As is shown in Figure 2, the results indicate that the interaction variable composed of social media exposure and source credibility of official media was significantly associated with risk perception ($\beta$ = −0.047, $p$ < 0.05, 95%CI = [−0.086, −0.004]), and the moderating effect was negative, which suggested that the greater the exposure to pandemic risk information through social media, the higher the risk perception. However, the high source credibility of official media can weaken the positive relationship between them and reduce the risk perception.

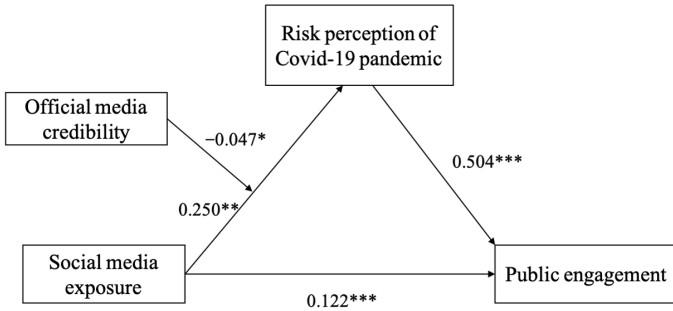

**Figure 2.** Results for the moderated effect of official media credibility. \* $p < 0.05$; \*\* $p < 0.01$; \*\*\* $p < 0.001$.

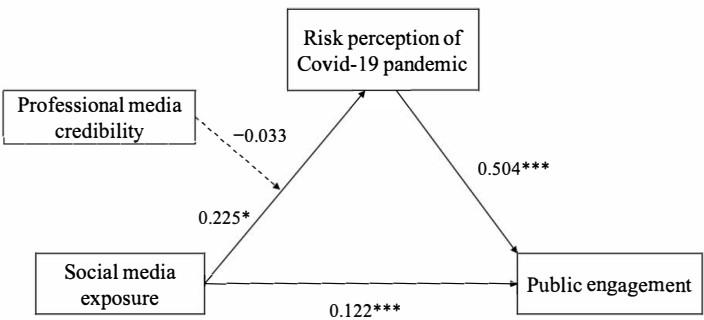

**Figure 3.** Results for the moderated effect of professional media credibility. \* $p < 0.05$; \*\*\* $p < 0.001$.

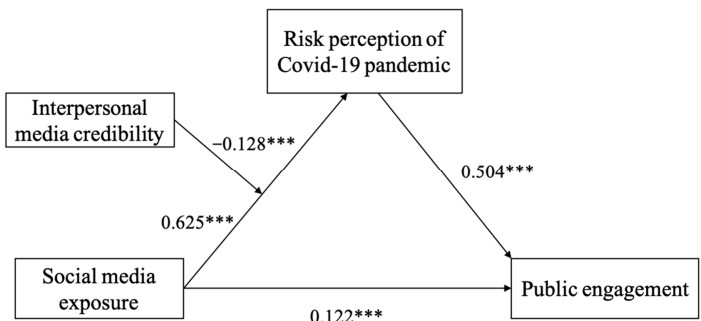

**Figure 4.** Results for the moderated effect of interpersonal media credibility. \*\*\* $p < 0.001$.

Figure 3 presents the interaction variable between the source credibility of professional media. Social media exposure did not significantly affect risk perception (β = −0.033, $p > 0.05$). According to Figure 4, the interaction variable between source credibility from interpersonal media and social media exposure negatively affected risk perception (β = −0.128, $p < 0.001$, 95%CI = [−0.158, −0.089]), indicating that high trust in interpersonal information sources from family and friends can also significantly weaken risk perception generated by social media and affect public risk engagement. Therefore, H2 is supported.

## 5. Discussion and Conclusions

Based on the Social Amplification of Risk Framework, this study constructs a mediated moderation model that includes social media exposure, source credibility, risk perception, and public engagement, expanding the social consequences of risk communication on social media in the context of health risks. The results show that the positive relationship between social media exposure and public engagement is mediated by risk perception and significantly moderated by source credibility. Specifically, both the credibility of official sources and interpersonal sources have a negative moderating effect, while the moderating effect from the source credibility of professional media is not significant. The amplification and mitigation effects of social media risks have been supported by empirical evidence.

These research findings have improved theoretical research on the risk amplification effects in a new media context and have led to proper risk communication strategies.

Many researchers have criticized that the SARF can only be examined through post-analysis of risks and cannot be empirically tested [52]. Our study empirically explores the relationship between social media exposure and public engagement, through which risk perception acts as a mediating factor. This is consistent with previous research findings in the context of the pandemic [54], emphasizing the psychological mechanisms involved in the social processing of risks.

This study provides empirical evidence for the moderating role of source credibility between media exposure and public engagement in health risk crises. Previous research has questioned the effectiveness of the positive relationship between risk information exposure and public participation, and quantitative studies have proposed other interfering factors such as social networks and interpersonal discussions [55,56]. This study further discovers that increased perceived credibility in official and interpersonal information can weaken the risk perception caused by social media exposure. Source credibility from interpersonal communication also provided psychological support to individuals, as previous research has shown the psychological benefits of interpersonal communication on social media during the pandemic [57]. In China's media institutional context, official media operate under a dual logic of information communication and public reassurance. As part of the media censorship system, official discourse is shaped by both top-down guidance and the strategic management of visibility. In the context of risk events, the release of authoritative information is centrally organized by the central government and the highest-level local authorities, which not only discloses the current severity of public health threats but also aims to tell emotionally resonant stories that unify and mobilize the public in support of the anti-pandemic effort. This selective visibility mechanism allows certain risks to be made visible while suppressing others, thereby constructing a socially constructed perception of the crisis reality. Thus, official media adopted a cautious and conservative communication strategy, balancing transparency with the imperative to avoid mass panic. This operation of "controlled credibility" functioned not merely to inform, but to shape how the public interpreted the severity, controllability, and social impact of the pandemic. Consequently, risk perception becomes a mediated outcome of both institutional trust and individual navigation of controlled information flows.

However, source credibility from professional media did not have a moderating effect. As new technology and methods of news consumption are developed, numerous studies over the past 30 years have shown a decline in trust in the media, as well as institutions and experts [58]. In the context of the COVID-19 pandemic, although high expectations were placed on experts and media from professional institutions, the absence of voices from professional medical media during the early stages of the pandemic did indeed impact people's perception and response to the course of the outbreak. Previous studies have shown that social media can provide an environment that fosters anti-intellectual attitudes, and divergent opinions among experts on health risks can create confusion and hesitation among the public to act [59,60]. Although society urgently needs health experts to intervene to improve risk communication, the expert system does not operate completely independently. The chaotic digital media public opinion environment and political pressure make the voices of health experts not clearly visible or even trusted by the public. The failure of professional media reflects the problem of the intervention mechanism in risk events. Professional advice is not supposed to be solely technically neutral, but to be validated through the public's rationality criteria in the political process.

In summary, mobilizing public engagement in health emergencies has always been an important issue in risk communication. Previous research has intensively focused on

media usage and neglected the characteristics of media information providers. This study demonstrates that as social media becomes increasingly involved in risk communication, it functions as a risk signal filter that can simultaneously amplify and attenuate public perceptions of risk. Notably, during the early stage of the pandemic, the high perceived credibility of official and interpersonal media sources weakened risk perception and, in turn, hindered collective risk actions—a finding that extends beyond the conclusions of previous SARF studies. SARF was originally developed to explain how social, institutional, and cultural processes amplify or attenuate risk signals in mass communication environments. However, social media platforms today introduce nonlinear, user-driven, and algorithm-mediated dynamics that are not fully accounted for in traditional SARF formulations. In these environments, risks may be amplified not only through interpersonal and institutional mechanisms, but also via platform algorithms that prioritize emotional content, sensationalism, and personalized engagement metrics. This process is underpinned by the logic of the attention economy, where user engagement serves as a key metric in the contest for visibility, and is further complicated by techno-political negotiations that determine what information is promoted and made visible within digital infrastructures. These shifts challenge the original assumptions of SARF that emphasized linear social pathways and stable institutional amplifiers.

Therefore, future studies could consider hybridizing SARF with more recent models, such as algorithmic gatekeeping theory and affective publics, to better capture the structural and affective conditions of risk processing in digital contexts. Such integration could help explain phenomena like persistent misinformation, the role of virality over expertise, and the platform-driven visibility of certain risk narratives over others. By embedding SARF in this broader theoretical conversation, researchers can more effectively analyze how risk is socially constructed, emotionally resonant, and politically consequential in digital societies.

## 6. Implications and Limitations

The findings of this study have certain implications for policymakers. To begin with, social media has become the main channel for the public to obtain information about the COVID-19 pandemic, and social media exposure has a significantly positive relationship with risk perception and public engagement. Considering social networks have a powerful influence on health decisions, the government, professional media, and experts should take the initiative to communicate with the public through social media [61]. Previous scholars proposed that the principle of frame engagement should be followed in public health risk communication: listening to dialog on social media, participating in the dialog, providing timely feedback, and promoting interaction and cooperation among all parties on health and public welfare issues [62]. Although risks are objective, the socially constructed risks mediated by the media still need to be regulated to maintain social stability.

Secondly, a high source credibility from official media and interpersonal networks can weaken risk perception. A prior study has shown that to enhance government credibility in risk communication, it is necessary to increase public perception of commitment to the government [63]. As a highly influential infectious disease, COVID-19 tremendously impacts individuals. The government should take the initiative to communicate with the public and release information correctly and immediately to establish an information authority. Meanwhile, we must not overlook the role of interpersonal communication in providing social support, which is crucial in alleviating people's fear of unfamiliar infectious diseases.

There are inevitably some limitations to this study. The research was conducted in 2020, during the early outbreak of COVID-19 in Hubei. Given that five years have passed, although the timeliness of the data presents a limitation, it remains necessary to reflect

on risk communication in the initial stages of a public health crisis and to undertake diachronic comparative studies. It is noteworthy that the pandemic is a dynamic process, which can have a long-term influence on public cognition of risk perception and source credibility. The cross-sectional data in this study cannot respond to causal effects, so additional attention needs to be paid to the external validity of the conclusions. Future studies may use longitudinal surveys for dynamic analysis. In addition, the sample includes a gender imbalance, with 69.2% of respondents being male, which may influence trust, risk perception, and engagement behaviors, as these can be shaped by gendered experiences in online environments. Future research should consider more balanced sampling strategies to ensure broader representativeness. Moreover, risk perception is an important psychological mechanism of the influence of media information exposure on public engagement, but the cognition of risk crisis is a multilayered process. To better understand social media's role in risk communication and the construction of a risk society, inter-disciplinary studies need to be explored by drawing more appropriate variables from political science, sociology, and other related disciplines.

**Supplementary Materials:** The following supporting information can be downloaded at https://www.mdpi.com/article/10.3390/covid5060084/s1, File S1: questionnaire table.

**Author Contributions:** Conceptualization, L.L.; methodology, L.L., and R.F.; data analysis, L.L.; writing—original draft preparation, L.L.; writing—review and editing, L.L. and R.F. All authors have read and agreed to the published version of the manuscript.

**Funding:** This research received no external funding.

**Institutional Review Board Statement:** This study was conducted in accordance with the Declaration of Helsinki, and approved by the Institutional Review Board of Shanghai Jiao Tong University (protocol code: H2020217I). The authors were affiliated with Shanghai Jiao Tong University from 2019 to 2023.

**Informed Consent Statement:** Informed consent was obtained from all participants involved in this study.

**Data Availability Statement:** The datasets generated and analyzed during the current study are available from the corresponding author upon reasonable request.

**Conflicts of Interest:** The authors declare no conflicts of interest.

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
