# Peer review of "Rethinking the Impact of Social Media Exposure and Source Credibility on the Social Amplification of Risk and Public Engagement During the COVID-19 Pandemic"

_covid, doi:10.3390/covid5060084_

Round 1
Reviewer 1 Report
This article has strong potential to contribute to the literature on digital risk communication, especially through its empirical application of SARF in the Chinese pandemic context. However, for publication, the manuscript requires revisions to acknowledge temporal limitations, shift the conclusion from citation-based to insight-driven, and update and integrate its reference base with more contemporary sources.
Major Comments
Lines 52–54: While the study's objectives are described, the manuscript lacks clear, explicitly formulated research questions. These should be clearly stated early in the introduction to structure the flow of argumentation and empirical design.
Theoretical Framing of SARF. Lines 62–115: Your paper makes good use of SARF but tends to summarize it descriptively rather than critically engaging with its assumptions and limitations in the digital age. It would be beneficial to elaborate how SARF accounts for non-linear, algorithmic amplification and information distortion in the context of social media ecosystems, especially within a semi-authoritarian media system such as China.
Temporal Validity and Relevance. The study was conducted in March 2020 during the early COVID-19 outbreak in Hubei. Given that it is now 2025, the temporal relevance of the data is a concern. The media environment, public attitudes, and information ecosystems have undergone significant transformation since then. This limitation should be explicitly acknowledged in the manuscript (e.g., in Lines 400–409) as it affects the generalizability and contemporary relevance of the findings.
Lines 304–377: The conclusion section still contains multiple references (e.g., [55]–[64]) when it should instead synthesize the study’s own empirical findings. A robust conclusion must focus on directly answering the research questions and highlighting theoretical and practical contributions derived from your data—not external citations.
Lines 291–293; 351–359: The non-significant moderating role of professional media credibility is not sufficiently discussed. This finding could be particularly interesting, given the expectation that expert sources would play a key role during health crises. Consider discussing the implications of this null result in light of declining trust in institutional expertise, especially in politicized or state-controlled contexts.
Lines 343–349: Although the paper mentions censorship and official discourse, it does not deeply explore how state control and selective visibility of information might influence both source credibility and risk perception. This aspect is particularly important in interpreting the role of official and interpersonal trust in Chinese society and media.
Reference Evaluation
Your reference list is generally well-aligned with the themes of SARF, social media, source credibility, and risk perception. Notable strengths include:
- Effective use of foundational SARF literature (e.g., [14], [54], [57]).
- Contemporary references on COVID-19 risk perception ([30], [34], [35]).
- Useful citations on source credibility in health communication ([18], [20], [43], [46]).
However, several areas need improvement:
- Some foundational sources (e.g., [15] Gerbner, 1976; [39] Hovland, 1953) are outdated and not sufficiently integrated with your current empirical focus. These could be replaced or supplemented with recent scholarship that directly addresses social media dynamics in risk communication post-2020.
- Several references (e.g., [1] Couldry, [3] Löfstedt, [19] Castells) are cited only briefly and lack integration into your conceptual or analytic framework.
- More recent literature (2022–2024) on social media misinformation, digital trust dynamics, and COVID-19 engagement patterns—especially those reflecting shifts in user behavior and platform policies—would strengthen the paper’s relevance and contribution.
Suggested additions:
- Limbu, Y. B., & Gautam, R. K. (2023). The determinants of COVID-19 vaccination intention: a meta-review. Frontiers in Public Health, 11, 1162861.
- Tsao, S. F., Chen, H., Tisseverasinghe, T., Yang, Y., Li, L., & Butt, Z. A. (2021). What social media told us in the time of COVID-19: a scoping review. The Lancet Digital Health, 3(3), e175-e194.
Minor Comments
- Abstract (Lines 9–23): Please mention the use of bootstrapped moderated mediation analysis and the survey location (Hubei, China) to give readers a better overview of the study design.
- Line 232: Clarify whether trust in source credibility was treated as a unidimensional or multidimensional construct in your model.
- Figures 2–4: Consider enriching the accompanying narrative interpretations of these graphs to clarify interaction effects for readers unfamiliar with moderated mediation models.
Reviewer 2 Report
Thank you for your efforts. Please:
1- Reflect the impact of political stability and political regime on level of public trust
2- Shed light on social media utilization in China
3- Please differentiate between urban and rural communities on perception of risk communication
None
Round 2
Reviewer 1 Report
Dear Authors and Editor.
Although I find the revised manuscript significantly improved and ready for publication, I would like to offer several optional suggestions that may help enhance the paper’s theoretical depth and broader academic contribution, either in its final revision or for future related studies.
First, the manuscript applies the Social Amplification of Risk Framework (SARF) appropriately, and the authors have already expanded its discussion in relation to digital media. However, a more reflective engagement on whether SARF remains conceptually sufficient in today’s algorithmically-curated, participatory media ecosystems could add further theoretical value. For instance, a brief reflection on the need to adapt or hybridize SARF with newer models of digital risk processing would be welcomed.
Second, the dataset includes a clear gender imbalance, with 69.2% of respondents being male. While this is mentioned in the methods, it would strengthen the paper’s transparency to briefly acknowledge this as a limitation—especially given that trust, risk perception, and engagement behaviors can be shaped by gendered experiences in online spaces.
Third, although the authors have incorporated recent references (2022–2024), a more explicit discussion of how public trust, media behaviors, and platform dynamics have shifted since the early pandemic would increase the contemporary relevance of the findings. For instance, user migration to short-video platforms and the decline in trust in mainstream sources could be mentioned in the implications.
Fourth, the conclusion could be slightly refined in tone. Some sentences (e.g., recommendations directed at policymakers) might benefit from being framed in more reflective, academically neutral language—emphasizing the importance of balancing transparency with institutional credibility, rather than instructive policy reminders.
Lastly, while Figures 2–4 effectively illustrate interaction effects, the authors may consider including a synthesized visual diagram or model summarizing how their empirical findings refine or expand the SARF. A visual summary could enhance the manuscript's clarity and help communicate its theoretical contribution more vividly.
These are optional points and not required for publication. However, I hope they are useful for further strengthening this meaningful and well-executed study.
Thankyou.
-
